# The Impact of School Closures during COVID-19 Lockdown on Visual–Motor Integration and Block Design Performance: A Comparison of Two Cohorts of Preschool Children

**DOI:** 10.3390/children10060930

**Published:** 2023-05-24

**Authors:** Mohd Izzuddin Hairol, Mahadir Ahmad, Muhammad Aminuddin Muhammad Zihni, Nur Fatin Syazana Saidon, Naufal Nordin, Masne Kadar

**Affiliations:** 1Centre for Community Health Studies (ReaCH), Faculty of Health Sciences, Universiti Kebangsaan Malaysia Jalan Raja Muda Abdul Aziz, Kuala Lumpur 50300, Malaysia; mahadir@ukm.edu.my (M.A.); naufalnordin55@yahoo.com (N.N.); 2Optometry & Vision Science Program, Faculty of Health Sciences, Universiti Kebangsaan Malaysia Jalan Raja Muda Abdul Aziz, Kuala Lumpur 50300, Malaysia; a168748@siswa.ukm.edu.my (M.A.M.Z.); a169221@siswa.ukm.edu.my (N.F.S.S.); 3Clinical Psychology & Health Behaviour Program, Faculty of Health Sciences, Universiti Kebangsaan Malaysia Jalan Raja Muda Abdul Aziz, Kuala Lumpur 50300, Malaysia; 4Centre for Rehabilitation & Special Needs Studies (iCaRehab), Faculty of Health Sciences, Universiti Kebangsaan Malaysia Jalan Raja Muda Abdul Aziz, Kuala Lumpur 50300, Malaysia; masne_kadar@ukm.edu.my; 5Occupational Therapy Program, Faculty of Health Sciences, Universiti Kebangsaan Malaysia Jalan Raja Muda Abdul Aziz, Kuala Lumpur 50300, Malaysia

**Keywords:** preschool, COVID-19 lockdowns, visual–motor integration, Block Design Test

## Abstract

The COVID-19 outbreak has led to the closure of educational institutions, which may prevent children from attaining skills essential for learning, such as visual–motor integration (VMI) and visuospatial constructional ability (often reflected with the Block Design Test, BDT). This study compares VMI and BDT performance between a pre-pandemic cohort (children who attended preschool in late 2019) and a post-pandemic cohort (those physically attending preschool for the first time at the end of 2021). Participants were children attending government preschools with similar syllabi catered for low-income families. The pre-pandemic cohort was part of an earlier study (n = 202 for VMI and n = 220 for BDT) before lockdowns commenced in March 2020. The post-pandemic cohort comprised 197 children who completed the Beery-VMI and 93 children who completed the BDT. Compared to the pre-pandemic cohort, the post-pandemic cohort had significantly lower mean Beery-VMI scores (t(397) = 3.054, *p* = 0.002) and was 3.162-times more likely to have a below average Beery-VMI score (OR = 3.162 (95% CI 1.349, 7.411)). The post-pandemic cohort also had significantly lower BDT scores than the pre-pandemic cohort (t(311) = −5.866, *p* < 0.001). In conclusion, children with disrupted conventional preschool education due to the COVID-19 lockdowns were more likely to have below-average VMI and lower BDT scores.

## 1. Introduction

The recent coronavirus disease 2019 (COVID-19) pandemic has impacted the educational attainment of more than 1.5 billion children worldwide [1]. Past epidemic events in history, such as severe acute respiratory syndrome (SARS) and N1H1, have also negatively impacted the normal development of children, including their educational attainment [2]. During the previous outbreaks of SARS and the N1H1 influenza pandemic, school closure strategies were implemented as a measure to contain the infection in children [3,4,5]. However, unlike the COVID-19 pandemic, these past epidemics led to restricted access to social and educational institutions only for affected schools, classes, or year groups, and in localized areas where they were most spread, such as in Taiwan [3], Singapore [4], and Beijing [5]. Meanwhile, schools in areas less affected by these epidemics, such as Malaysia, remained open.

During the COVID-19 pandemic, educational institutions in Malaysia, as in the rest of the world, were closed as one of the measures implemented to minimize the transmission of the severe acute respiratory syndrome coronavirus 2 (SARS-CoV-2). All schools, preschools, and daycare centers were closed as the Movement Control Order (MCO) was imposed on 18 March 2020. A series of school reopening and reclosing procedures followed in response to subsequent waves of the virus until January 2021, further disrupting children’s learning process. From 1 March 2021, Malaysian children returned to school in several phases, with priorities given to those sitting for national examinations [6]. Restrictions were also applied particularly to preschool opening, including alternate day attendance, increased social distancing, compulsory daily temperature checks, and reduced class size [6,7]. The pandemic also changed the typical school period; during pre-pandemic, the school year began in January and ended in November. Since 2021, the school year has started in March and has ended in February the following year.

Previous studies have reported the detrimental consequences of prolonged restricted access to educational institutions on children’s health and social well-being. Evidence of learning losses has been reported during routine school closures, such as the typical summer break. Younger children with disrupted attendance to kindergarten and preschool were reported to have lower academic achievement and executive function skills [8,9,10]. Specifically for the COVID-19 pandemic, the associated lockdowns and movement restrictions are thought to have contributed to a learning loss of between 0.3 to 1.1 years of schooling [11]. In addition, there was an estimated 30% increase in achievement gaps between those from different socioeconomic backgrounds [12]. Children in post-COVID-19 cohorts also have reduced math scores [13,14], language [14], and losses in motor and cognitive development [15]. Lockdowns led to increased sedentary screen time [16], where more time at home led to increased TV, PC, tablet, and mobile phone viewing time of up to four hours a day [17]. There are also reports on increased childhood obesity prevalence [18], which was suggested to be associated with increased screen exposure due to changes in diet and snacking [19]. The implemented control measures also contributed to mental health deterioration among children [20]. These effects may be more significant for preschool-age children transitioning into school entry during the pandemic [2].

The preschool years are critical for children, as access to early learning has impacts that are continuous across children’s life spans, affecting educational achievement, delinquency, and earning potential as adults [21]. As such, the skills they develop in this period are crucial before formal schooling starts. One of the essential abilities to be acquired during the preschool years is visual motor skills [22]. The degree to which visual perception and motor coordination are closely synchronized is measured by visual motor integration (VMI) [23]. VMI is associated with learning the mastery of handwriting [24]; thus, children with VMI deficits have an inadequate spatial organization of written language [25], lower reading performance [26], and lower academic achievements [27]. Thus, deficits in VMI during preschool years could affect these children’s learning experience when they transition into primary school.

Another skill of interest that is also crucial is children’s visuospatial constructional ability, which involves the ability to perceive objects or diagrams as a set of parts and then use the parts to construct a replica of the original [28]. Visuospatial constructional ability is often reflected with the Block Design Test (BDT) [29]. In BDT, participants use small blocks to reproduce patterns, where the process involves the analysis, synthesis, and manipulation of visual stimuli [30]. As a measure of visuospatial abilities, young children’s BDT performance is closely related to mathematical achievements [30,31]. Visuospatial ability is also a significant predictor of preschool children’s mathematics ability when they join primary school [32,33]. In addition, BDT performance is regarded as a good predictor of general intellectual ability [34]. Thus, deficits in the ability to visualize and spatially construct objects, as measured with the BDT, could adversely affect young children’s future academic achievements.

Strict restrictions for physical access to early education institutions may prevent children from attaining these skills essential for learning, particularly those of lower socioeconomic backgrounds. A previous study has reported that preschool children from lower-income families have poorer VMI performance than those from higher-income families [35]. Children from this disadvantaged group were more likely to be impacted by their inability to attend physical classes for formal learning. They would also likely be restricted from online learning during the COVID-19 pandemic lockdowns. In this study, we asked whether their VMI skills and BDT performance were more negatively impacted compared to their peers who had the opportunity to attend preschool lessons physically before the pandemic began.

Thus, this study compares VMI and Block Design Test performance between two cohorts of Malaysian children attending preschools catered to low-income families. The pre-pandemic cohort was a sample of children before preschool closures were implemented (between 2019 and early 2020). The post-pandemic cohort consisted of those who attended preschools for the first time after physical teaching and learning recommenced in Malaysia towards the end of 2021.

## 2. Materials and Methods

### 2.1. Participants

Participants were children between the ages of 5 and 6 attending KEMAS preschools in Klang Valley in Peninsular Malaysia. KEMAS preschools are governed by the Community Development Departments (with the Malay acronym, JKM), catering to children from very low-income families in suburban and rural areas [36]. Therefore, all participants were approximately homogenous regarding their socioeconomic background and the preschool syllabus they received.

The pre-pandemic cohort was part of an earlier study [35] that measured VMI and BDT performance between November 2019 and January 2020. In that study, the participants were 435 children attending KEMAS and private preschools. For the current study, only the data of children attending KEMAS preschools were extracted. The extracted data, consisting of around 200 individuals, would give the study 90% power to detect an effect size of 0.3.

The post-pandemic cohort were children attending KEMAS preschools in-person for the first time since the Movement Control Order restrictions were lifted. Data were collected between November 2021 and March 2022. As with the earlier pre-pandemic cohort study, the JKM of the state of Selangor and the Federal Territory of Kuala Lumpur provided the complete list of the state-funded preschools. Using simple random sampling, the preschools were chosen from the list. The preschools were contacted after receiving authorization from the research ethics committee and the preschools’ principals. Prior to data collection, the parents or legal guardians of the children provided written informed consent. Potential child participants who refused to agree or if their parents refused to provide consent were excluded. Children were also excluded if the parents provided information on any disabilities. Vision screening sessions were conducted to ensure that participants from both pre-pandemic and post-pandemic cohorts had distance and near visual acuities equal to or better than 0.3 logMAR (logarithms of the Minimum Angle of Resolution) or better in each eye (equivalent to Snellen 6/9) with stereopsis of 170 s of arc or better.

The Research Ethics Committee of Universiti Kebangsaan Malaysia (UKM/PPI/800-1/1/5/JEP-2019-476 and UKM/PPI/111/8/JEP-2021-172) and the Malaysian Ministry of Rural Development (KEMAS.BPAK620-02/01/01 Jld 15 (2) and KEMAS.BPAK620-02/01/01 Jld 20 (55)) both approved the study’s ethical conduct. The principles of the Declaration of Helsinki were adhered to in every element of this study’s conduct.

### 2.2. Measurements

The performance for visual–motor integration was determined with the Beery-Buktenica Developmental Test of Visual Motor Integration (Beery-VMI) (6th edition). The norm-referenced instrument is widely used as a screening tool to identify children who have not fully integrated their motor and visual abilities [23]. The test requires participants to copy a series of up to 30 geometric shapes using pencil and paper and is suitable for children from 2 years old. The test is reported to have high reliability, with an inter-rater reliability between 0.92 to 0.98 and a test–retest reliability correlation of 0.92 for a 2-week interval in adults and children [23,37,38].

The Block Design Test (BDT) is a subtest included in the Wechsler Preschool and Primary Scale of Intelligence-Fourth Edition (WPPSI-IV) [39], suitable for children ages 2 years 6 months to 7 years 7 months, and the Wechsler Intelligence Scale for Children-Fifth Edition (WISC-V), suitable for children ages 6 years 0 month to 16 years 11 months [40]. The BDT consists of two-dimensional pictures/designs printed in the Stimulus Book, where a child views the picture and uses two-color blocks to re-create the design. According to the tests’ Technical and Interpretive Manuals, the BDT measures nonverbal concept formulation and reasoning, simultaneous processing, visual–motor coordination, learning, and the capacity to distinguish between figures and the ground in visual stimuli. It was created to assess children’s capacity to analyze and synthesize abstract visual data [40]. The BDTs from both WPPSI-IV and the WISC-V have a good subtest reliability coefficient, with high correlations between them [40].

### 2.3. Procedures

The measurements of the VMI scores with the Beery-VMI were conducted in preschool classrooms under sufficient room lighting. Participants wore their existing optical prescription if required. Participants were presented with the Beery-VMI booklet that contained printed geometric shapes. They were instructed to draw the shapes in the dedicated blank space provided using a pencil, and the use of an eraser was not allowed, following the test’s standard instructions. Author NN, a doctoral graduate with three years of experience conducting the test, administered the test to each child individually. NN scored the accuracy of the according to the Beery-VMI scoring criteria, and this was confirmed by author MK, an occupational therapist with more than 17 years of experience working with young children. Discrepancies in scoring were agreed upon by consensus by NN and MK. Raw scores were converted to standard scores based on the instruction manual [23]. In the manual, Beery-VMI standard scores were categorized into Very High (score > 129), High (120–129), Above Average (110–119), Average (90–109), Below Average (80–89), Low (0–79), and Very Low (<70).

For the Block Design Test, the BDT from WPPSI-IV was used for the 5-year-old participants, and BDT from WISC-V was used for the 6-year-old participants [39]. Participants were shown printed two-dimensional red-and-white designs displayed on the workbook using the standard presentation procedures outlined in the Administration and Scoring Manuals. Then, they were instructed to reproduce the designs by arranging cubes with red, white, half-red and half-white faces. The test was administered to each participant individually in the preschool classrooms. Author MA, a registered clinical psychologist with over 12 years of experience, supervised the BDT administration with NN as the assistant. Each trial was timed, and raw scores were calculated as specified in the manuals. All participants completed the test in a single session that lasted approximately 10 min. The raw scores ranged from 0 to a maximum of 58 for WISC-V and 0 to a maximum of 34 for the WPPSI-IV. The total BDT raw scores were converted into scaled scores using the age-appropriate normative tables in the Administration and Scoring Manual for both tests [39,40] that allowed comparison of the children’s performance among their same-age peers.

### 2.4. Statistical Analysis

Descriptive statistics were used to analyze all continuous quantitative variables (participant’s age, Beery-VMI scores and BDT scores). Independent sample *t*-tests were used to compare the means for Beery-VMI scores and BDT scores between gender and between participant cohorts. Additionally, the Beery-VMI scores were also re-categorized and modified from the Beery-VMI 6th Edition Manual [23]. Scores 89 and lower were labelled ‘Below Average’. Scores between 90 and 109 were labelled ‘Average’, and those 110 and higher were labelled ‘Above Average’. The association between the participant cohort category and the Beery-VMI score category was examined using the chi-square test. Multinomial logistic regression analyses were conducted to determine odds ratios (OR) where the OR represented the probability of the participant obtaining Beery-VMI scores that were in the ‘Below Average’ and ‘Average’ categories, with the ‘Above Average’ Beery-VMI score category as the reference. The first regression model was unadjusted, while the second model was adjusted by the participants’ age. For all statistical tests, the significance level α was set at 0.05. All data were sorted in MS Excel, and statistical analyses were conducted using IBM’s Statistical Package for Social Sciences (SPSS) version 22.

## 3. Results

### 3.1. Beery-VMI Performance

Two hundred and two children (mean age: 5.81 ± 0.45 years) completed the Beery-VMI before preschool closures following the Movement Control Order (MCO) implementation. When preschool reopened in 2021, 197 children completed the Beery-VMI (mean age: 5.94 ± 0.57 years). Compared to the post-pandemic participants, the pre-pandemic participants were significantly younger (t(397) = −2.557, *p* = 0.011) with a small effect size (Cohen’s d = 0.253). In both cohorts, there were slightly more males than females. Table 1 summarizes the characteristics of the study participants based on age and sex.

Figure 1A shows the mean Beery-VMI scores by sex and cohort. For the pre-pandemic cohort, the mean Beery-VMI score for males (98.02 ± 9.08) is slightly lower than those for females (99.90 ± 9.21), but they are not different statistically (t(200) = −1.450, *p* = 0.149). The overall pre-pandemic cohort’s mean Beery-VMI score is 98.84 ± 9.16.

For the post-pandemic cohort, the mean Beery-VMI scores for males (96.46 ± 8.79) and females (95.65 ± 8.74) are also not significantly different (t(196) = 0.647, *p* = 0.519). Despite the post-pandemic cohort being older, the mean Beery-VMI score for the post-pandemic cohort combined across sex (96.10 ± 8.75) is significantly lower than that measured in the pre-pandemic cohort (98.84 ± 9.16) (t(397) = 3.054, *p* = 0.002), with an effect size between small to medium (Cohen’s d = 0.306). However, both cohorts’ mean Beery-VMI scores are within the score range for the Average category. 

Figure 1B compares the Beery-VMI scores presented in the Below Average, Average, and Above Average categories. It is found that 13.86% (n = 28) of children in the pre-pandemic cohort obtained a Below Average VMI (scores less than 90); the percentage increases to 22.34% in the post-pandemic cohort. In addition, 12.38% of the sample from the pre-pandemic cohort obtained an above average VMI (scores higher than 110); the proportion reduced to 6.09% for the post-pandemic cohort. The relationship between the participant cohort and VMI performance is statistically significant (Χ^2^ (2, N = 399) = 8.282, *p* = 0.016). That is, compared to the pre-pandemic cohort, the post-pandemic cohort is likelier to have a Beery-VMI score that is below average. They are also less likely to obtain a Beery-VMI score that is above average.

Table 2 shows the results of the multinomial logistic regression. The age-adjusted regression model reveals that preschool children in the post-pandemic cohort were 3.162 times likelier to get a Beery-VMI score in the below average category (OR = 3.162 (95% CI 1.349, 7.411)) than the children in the pre-pandemic cohort.

### 3.2. Block Design Test performance

Table 1 also shows the distribution of participants from the two cohorts who completed the BDT. There are 220 children in the pre-pandemic cohort, with a mean age of 5.78 ± 0.45 years. In the post-pandemic cohort, there are 93 children with a mean age of 6.13 ± 0.53 years. The post-pandemic cohort is significantly older than the pre-pandemic cohort (t(311)= −5.87, *p* < 0.001, Cohen’s d = 0.726).

For the pre-pandemic cohort, the BDT scores for males (9.04 ± 2.89) and females (9.10 ± 2.57) are not significantly different (t(218)= −0.16, *p* = 0.873). The overall mean BDT score for the pre-pandemic cohort is 9.07 ± 2.74.

For the post-pandemic cohort, the BDT scores for males (6.86 ± 3.68) and females (7.40 ± 3.02) are also not significantly different (t(91) = −0.758, *p* = 0.450). Combined across sex, the BDT score for the post-pandemic cohort was 7.11 ± 3.39, which is significantly lower than that obtained by the pre-pandemic cohort (9.07 ± 2.74) (t(311) = 5.377, *p* < 0.001, Cohen’s d = 0.665), with a medium effect size. That is, the lower BDT scores measured in children who experienced restricted access to preschool education during the COVID-19 lockdowns than their pre-pandemic peers are both statistically and practically significant.

## 4. Discussion

The present work compared the Beery-VMI and the Block Design Test performances between two cohorts of preschool children who attended similar preschool types and with similar socioeconomic backgrounds. The pre-pandemic cohort scored higher for both the Beery-VMI and BDT than the post-pandemic cohort, despite the latter’s mean age being slightly older.

Although both cohorts’ mean Beery-VMI scores fell within the Average range, the post-pandemic cohort had greater than three times the risk of having a below-average VMI performance compared to the pre-pandemic cohort. Our findings with the Beery-VMI performance suggested important implications. Firstly, looking only at the mean Beery-VMI score would be insufficient to fully reveal the impact of school closures due to the COVID-19 lockdowns on preschool children’s visual–motor integration abilities. Our results also add to the evidence that the implementation of school closure strategies during the COVID-19 pandemic has had a broad range of impacts on children’s development, in addition to the economic costs and social interruptions [41].

The deterioration in VMI, particularly among preschool children, is of concern as VMI is a predictor of handwriting performance [42,43] and handwriting readiness [44,45], with possible further implications on their school readiness. The results add to the evidence in the literature that restricted access to formal learning due to COVID-19 lockdowns detrimentally affected young children’s growth and academic-related progress, as children struggle to accept new learning modes [46]. The changes reported in children due to the lockdowns include motor and cognitive development and attitudes toward learning [15]. A recent study also reported that motor tasks and pre-literate abilities were detrimentally affected in Portuguese children by the COVID-19 lockdowns, where skills differences increased between children with higher and lower socioeconomic status [47].

The need for quick adaptation to novel but unpredictable learning methods during the COVID-19 lockdowns [47] could have negatively affected the development of visual–motor integration skills in young children [48,49]. School closures have accelerated the use of tablets as a medium for online learning [50], partly responsible for replacing traditional pencil-and-paper activities. It has been reported that young children who use a tablet for multiple hours per week have lower fine motor and visual perception abilities than those who do not or spend fewer hours with tablets [48]. Extensive tablet use also affects fine motor precision, fine motor integration, and manual dexterity [49]. In addition, a recent study showed that online learning has similar effects to those obtained when no formal learning is conducted during the usual summer school breaks [13], suggesting that online student learning is ineffective [51]. Although the use of digital technology may support self-directed learning in young children [52], its consequences on the development of fine motor skills, which are essential for various tasks such as handwriting, must be considered.

Other than the loss of formal schooling in physical classrooms, children educational attainment were also impacted in different ways, including losing access to crucial services such as daycare and health services [53]. Particularly for children from low-income families, these include the exacerbation of consumption of unhealthy diets [54] that is associated with lower academic performance [55,56]. Children from low-income families are also more likely to fail vision screening, indicating the presence of associated uncorrected refractive error and blurred vision [57]. Previous studies have shown that blurred vision, especially at near range, negatively impacts VMI performance in young children [58], reading and visual information processing during sustained near work [59], and academic performance [60].

The lockdowns associated with the COVID-19 pandemic have also negatively affected children’s BDT scores, adding to similar reports for their mental health [20], obesity prevalence [18], and cognitive development [15]. The lockdowns would have restricted access to the materials usually provided in physical preschool classrooms needed for the children to develop their visual–spatial abilities. As it has been suggested that guided block play could enhance preschool children’s mathematical skills [61], the lower BDT performance, as a measure of visuospatial abilities, would predict lower mathematical achievements and later quantitative ability [30,31]. Thus, children without sufficient education access during the COVID-19 lockdown period might have poor learning outcomes [51,62], worsening their visuospatial ability development.

Our study did not find a significant difference in BDT performance between males and females. The similarities in BDT performance between males and females found in this study are similar to those reported earlier, which also employed similar tests [63,64]. Although there is evidence for a male advantage in visual–spatial abilities [65], the lockdowns likely impacted children’s visual–spatial skills similarly regardless of gender. Instead, the development of these abilities might be more affected in children from low-income families, such as those in the current study.

Measures were taken by educational authorities to ensure that learning could resume during the pandemic, where teachers and children had to adapt to online teaching and learning [50]. Online and home-schooling implemented during the COVID-19 pandemic became more difficult and less attainable, particularly for those who were vulnerable, such as children of undocumented migrants [62], younger children [66], and those with low socioeconomic backgrounds [15,67]. These children might not have benefited from the sudden change in the new educational delivery method, as some families were unable to provide their children with appropriate devices, data, and connectivity [62,68]. They were also less likely to have sufficient learning time [69] and to receive adequate assistance for learning [66]. Lower-income households are also less likely to have appropriate educational materials, with parents that are less likely to be able to help children in their learning process [70]. In combination, these factors made the children struggle to complete given homework and achieve the intended learning outcomes [54]. These circumstances further hindered their access to activities that might have helped them to develop their visual–motor and visual–spatial abilities during the lockdown period. Thus, policymakers should act to mitigate the adverse effects of the lockdowns on these children to ensure that they catch up after they resume formal schooling.

In addition, the lockdowns have also led to increased internalizing and externalizing symptoms in children [71,72], which also affected parents with an associated increase in parental stress [72,73]. Future studies could evaluate if the affected developmental processes in children, as reported in this study, also lead to increased parental stress.

The study has several limitations. First, the study was conducted in preschools catering to low-income families, where class size was usually large. Although the children performed all tests in a secluded corner to minimize distractions, it was impossible to completely eliminate the noise made by other children, which could have affected the attention of the tested child. Secondly, all children were from low-income families and resided in semi-urban areas. Those from rural areas with different socioeconomic backgrounds with different opportunities and access to learning atmospheres might have had their VMI and BDT performance affected in different ways by the COVID-19 lockdowns. Lastly, the information on the participants’ neurophysiological profiles was obtained from their parents only. There could be possibilities that there might be children with undiagnosed conditions that could influence the study’s outcomes.

## 5. Conclusions

Children from the post-COVID-19 pandemic cohort, who attended preschools for the first time after physical teaching and learning resumed, were more likely to have below-average visual motor integration (VMI) performance and lower Block Design Test (BDT) scores than children who attended preschool before the pandemic began. The lower VMI and BDT performance was similar for both boys and girls. In addition, using the mean Beery-VMI score only is insufficient to fully reveal the impact of school closures due to the COVID-19 lockdowns on preschool children’s visual–motor integration abilities.

## Figures and Tables

**Figure 1 children-10-00930-f001:**
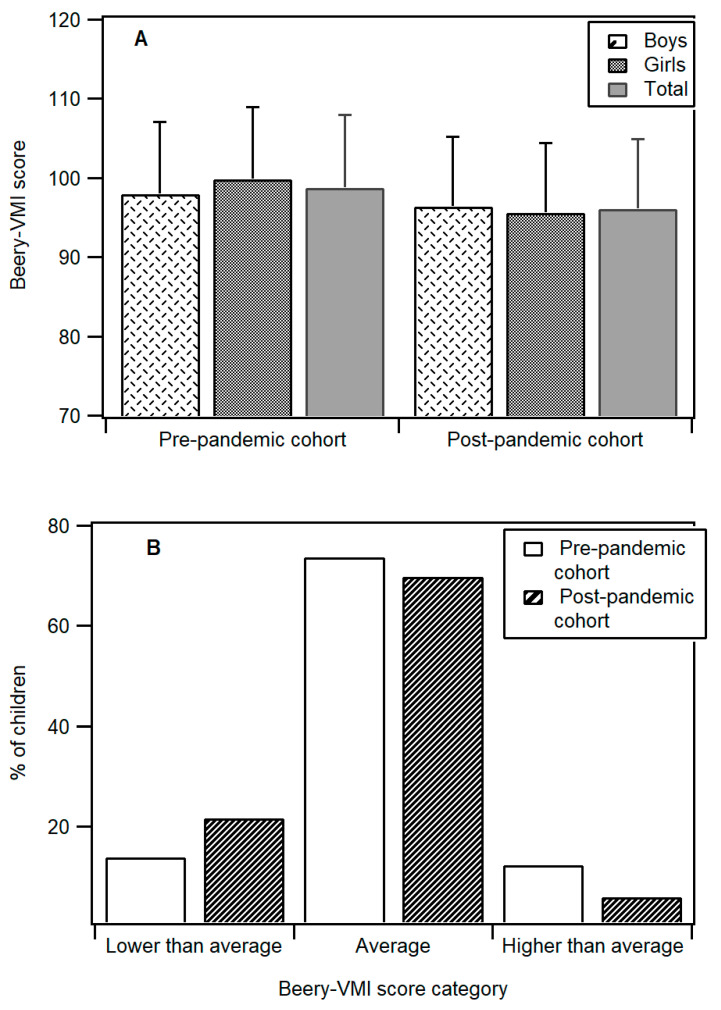
(**A**). Beery-VMI means for pre-pandemic and post-pandemic cohorts. (**B**). Beery-VMI score categorized into Below Average, Average, and Above Average for pre-pandemic and post-pandemic cohorts.

**Table 1 children-10-00930-t001:** Number and age of the study participants that completed the Beery-VMI and the Block Design Test.

	Pre-Pandemic Participants			Post-Pandemic Participants		
	Male (%)	Female (%)	Total	Male (%)	Female (%)	Total
**Beery-VMI**						
Number of participants, n	114 (56.44)	88 (43.56)	202	108 (54.82)	89 (54.18)	197
Mean age, year	5.78 ± 0.45	5.85 ± 0.46	5.81 ± 0.45	5.92 ± 0.59	5.97 ± 0.55	5.94 ± 0.57
**Block Design Test**						
Number of participants, n	121 (55.00)	99 (45.00)	220	50 (53.80)	43 (46.20)	93
Mean age, year	5.76 ± 0.45	5.81 ± 0.45	5.78 ± 0.45	6.08 ± 0.53	6.18 ± 0.53	6.13 ± 0.53

**Table 2 children-10-00930-t002:** Multinomial logistic regression analysis for Beery-VMI performance.

Cohort	Beery-VMI Score Category	Unadjusted Model			Adjusted Model ^1^		
		Odds Ratio	*p*-Value	95% CI (Lower–Upper)	Odds Ratio	*p*-Value	95% CI (Lower–Upper)
Post-pandemic ^2^	Below average	3.274 *	0.005	1.420–7.550	3.162 *	0.008	1.349–7.411
	Average	1.971	0.067	0.954–4.074	1.885	0.095	0.865–3.973
	Above average	1	-	-	1	-	-

^1^ Adjusted by age, ^2^ With pre-pandemic cohort as reference. * Significant at *p* < 0.05.

## Data Availability

Data are openly available at https://doi.org/10.6084/m9.figshare.22663975.

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
