# Peer review of "The Impact of School Closures during COVID-19 Lockdown on Visual–Motor Integration and Block Design Performance: A Comparison of Two Cohorts of Preschool Children"

_children, 2023, doi:10.3390/children10060930_

Round 1

Reviewer 1 Report

Line 67 - Replace 'writing' with 'learning the mastery of handwriting'

Line 68 - I am missing here the conclusion as it is given at the end of this paragraph regarding the BDT

Line 88-153 - Please make subparagraphs in the Material and Method section: 1) participants; 2) measurements; 3) procedure; and 4) Statistical analysis and make sure that the different information, which is now intertwined, is ordered according to the subparagraphs.

Line 119Replace Beery-VMI Developmental Test by Beery-Buktenica Developmental Test of Visual-Motor Integration (Beery-VMI) (6th edition)

Line 119-120 Give more information about the Beery-VMI: goal, procedure and psychometric properties

Line 120 Was the test administered as a class or individually per child? Why 15 minutes - there is no time pressure on this test. 

Line 130 - Categorized in what sort of scores?

Line 131-135 - I miss the information about the psychometric properties of the BDT

Line 143 - Be clearer about what kind of scores are used….

Line 215-217 - I don't understand what you want to say with this sentence, can you explain better what you mean?

Line 219 - the Beery-VMI is also a predictor of handwriting readiness: 1) van Hartingsveldt, M. J., Cup, E. H., de Groot, I. J., & Nijhuis‐van der Sanden, M. W. (2014). Writing Readiness Inventory Tool in Context (WRITIC): reliability and convergent validity. Australian occupational therapy journal, 61(2), 102-109. & 2) van Hartingsveldt, M. J., Cup, E. H., Hendriks, J. C., de Vries, L., de Groot, I. J., & Nijhuis-van der Sanden, M. W. (2015). Predictive validity of kindergarten assessments on handwriting readiness. Research in developmental disabilities, 36, 114-124.

Line 237 - Please add a reference

Line 328 - replace reference 17 by - Beery, K.E.; Beery, N.A. The Beery-Buktenica Developmental Test of Visual-Motor Integration: Administration, Scoring, and Teaching Manual, 6th ed.; NCS Pearson: Minneapolis, MN, USA, 2010

Author Response

We thank Reviewer 1 for the comments, which have helped us to improve the manuscript tremendously. Please find the point-by-point responses in the attached file. 

Reviewer 2 Report

The study is interesting but I have various considerations. As regards Materials and Methods, it should be more appropriate to make subparagraphs, for example one in which the participants are described, one referring to the tools and one for the measures of the statistical methods used.
The number of participants in the post-pandemic cohort should be specified. It would be interesting to have information also regarding the neuropsychological profile of these children, if someone has genetic syndromes, intellectual disabilities, psychomotor delays, neurodevelopment disorders, factors which could also influence the outcomes.
Furthermore, when the Beery-VMI Developmental Test is named, the reference citation should be included, at least a brief description of the test should be given, and the first time it should be spelled out completely "Beery-Buktenica Developmental Test of Visual-Motor Integration". In the Block Design Test (BDT) should be described the time trials and what skills it measures.
Finally, the bibliography could be expanded: for example, for possible future studies, it could also be evaluated the difference in other important factors, in emotional problems between pre- and post-pandemic or even if the changes found in children influence the level of parental stress. In fact, in one study, the COVID-19 pandemic and the corresponding measures adopted led to an increase in internalizing and externalizing symptoms in children and adolescents with neuropsychiatric disorder emerged; similarly, parental stress increased during COVID-19 and higher level of stress in parents can be related to the internalizing symptoms of their children.

English overall is fine.

Author Response

We thank Reviewer 2 for the comments. Please find the point-by-point responses in the attached file. 

Round 2

Reviewer 2 Report

The changes made are fine.

The literature could be further expanded: an Italian study "Impact of COVID-19 Pandemic on Children and Adolescents with Neuropsychiatric Disorders: Emotional/Behavioral Symptoms and Parental Stress" suggested that the COVID-19 pandemic and the corresponding measures adopted led to an increase in internalizing and externalizing symptoms in children and adolescents with neuropsychiatric disorder. Similarly, parental stress increased during COVID-19 and a higher level of stress in parents can be related to the internalizing symptoms of their children.